# Development of Traceable Mouse Models of Advanced and Metastatic Bladder Cancer

**DOI:** 10.3390/cancers16122245

**Published:** 2024-06-17

**Authors:** Emma Desponds, Konstantina Kioseoglou, Hana Zdimerova, Marco Ongaro, Grégory Verdeil, Marine M. Leblond

**Affiliations:** 1Department of Oncology, Lausanne University Hospital (CHUV), University of Lausanne, 1015 Lausanne, Switzerland; emma.desponds@unil.ch (E.D.); konstantina.kioseoglou@unil.ch (K.K.); hana.zdimerova@seznam.cz (H.Z.); marco.ongaro@unil.ch (M.O.); 2Ludwig Institute for Cancer Research, University of Lausanne, 1015 Lausanne, Switzerland

**Keywords:** bladder cancer, mouse model, metastasis

## Abstract

**Simple Summary:**

Poor prognosis for advanced bladder cancer underlines the need for a better understanding of tumor resistance to treatment in relevant preclinical models. We developed new models of inducible and orthotopic muscle-invasive bladder cancer, with strong phenotypical and genetic resemblances to the human disease. These models enable real-time tracking of tumor progression and metastasis through bioluminescence imaging to facilitates the testing of new therapies. This valuable tool is important for advancing bladder cancer research and developing more effective treatments.

**Abstract:**

Bladder cancer (BC) is the fourth most common cancer in men, with a poor patient prognosis for advanced disease. The poor survival of patients with muscle-invasive bladder cancer (MIBC) and metastatic status emphasizes the urgent need to develop new therapies. Lacking in the field of BC is the availability of relevant advanced BC mouse models, especially metastatic ones, that accurately recapitulate the complexities of human pathology to test and study new therapeutic strategies. Addressing this need, we developed a traceable mouse model of BC that expresses tumor-associated antigens within the context of advanced muscle-invasive BC. This novel system was achieved through the deletion of the *tp53* and *pten* genes, alongside the incorporation of the fusion construct of Firefly luciferase (Luc) and the SIYRYYGL (SIY) T-cell antigen. We validate that the presence of the transgene did not impact on the development of the tumors while allowing us to measure tumor progression by bioluminescence. We show that the transgene did not influence the composition of the immune tumor microenvironment. More importantly, we report that this model was unresponsive to anti-PD-1 treatment, as in the majority of patients with BC. We also develop a new model based on the orthotopic injection of BC clonal cell lines derived from our first model. We demonstrate that this new model invades the muscle layer and has a metastasis development rate of 83%. The advantage of this model is that we can visualize tumor growth and metastasis development in vivo. These mouse models’ characteristics, displaying many similarities with the human pathology, provide a valuable tool for tracking tumor progression, metastasis spread in vivo, and treatment resistance, as well as exploring fundamental and translational aspects of BC biology. This work contributes to the improvement in the landscape of mouse models of advanced BC for testing new therapeutic strategies.

## 1. Introduction

Bladder cancer (BC) is a significant global health challenge, with the World Health Organization (WHO) reporting approximately 600,000 new cases annually. At diagnosis, 75% of BC cases are classified as nonmuscle invasive BC (NMIBC), which is characterized by a high rates of tumor recurrence and progression [1]. Muscle-invasive BC (MIBC) accounts for 25% of cases and is associated with a 5-year survival rate of about 50%. Unfortunately, about half of patients with MIBC will develop distant metastasis, leading to a median survival of merely 15 months [2]. The use of immune checkpoint inhibitors (ICIs), especially anti-PD-1/PD-L1 therapy, has been approved by the Food and Drug Administration. However, about 80% of patients with advanced MIBC do not respond to this therapy [3], even though BC is a smoking-related cancer associated with a high mutational burden and tumor-associated antigens (TAAs) that should favor an ICI response [4]. This highlights the urgent need for an in-depth understanding of the immune mechanisms governing tumor control and therapy resistance but also the development of new immunotherapeutic strategies, tasks that necessitate preclinical investigations. 

The use of mouse models is crucial for the study of the biological behavior of cancer, as well as for testing the efficacy and toxicity of novel treatments in living organisms. They play a role that cannot be replaced, bridging cell-based experiments to clinical trials. The field of BC research suffers from a scarcity of relevant animal models that accurately reflect the complexity of human pathology, including tumor growth, histopathological features, and molecular characteristics. Animal models for BC research can be broadly categorized into autochthonous models, such as chemically induced and genetically engineered mouse models (GEMMs), as well as nonautochthonous systems, in which tumor cells are injected either heterotopically (i.e., outside the bladder) or orthotopically (i.e., within the bladder) [5]. While orthotopic models using MB49 bladder tumor cell catheterization is widely used for BC, they often exhibit rapid tumor growth kinetics, limiting the therapeutic window [6], and are poorly representative of the gene expression profile found in human urothelial tumors [7]. Another limitation of the MB49 model is the absence of tumor cell invasion through the muscle wall, restricting the use of this model for the NMIBC stage [8]. 

Carcinogen-induced MIBC models using N-butyl-N-(4-hydroxybutyl) nitrosamine (BBN) have been shown to replicate the high mutational burden described in human MIBC [9] and present a human-like tumor immune microenvironment [7]. However, as the carcinogen is delivered in the drinking water, its systemic effect makes it challenging to estimate the contribution of other tissues on tumor development [5]. Moreover, bladder tumor development usually obstructs ureters leading to hydronephrosis and death before metastases can arise [10]. BBN model is also known to induce the development of heterogenous tumor phenotypes and progressions, emphasizing the difficulty in using this model for preclinical studies [5].

A GEMM involves specific genetic manipulations to induce bladder tumorigenesis and has the particularity of being stable and reproduceable. One main difficulty in generating these models is the limited number of promoters displaying bladder-specific expressions [5]. To overcome this problem, the use of adenoviral vector encoding for the Cre-recombinase directly delivered into the bladder of mice with floxed tumor-suppressor genes has been developed [11,12,13,14,15]. Our laboratory has recently characterized, at the immune cell level, a Cre-LoxP-regulated mouse model of BC that involves the deletion of Tp53 and Pten genes [16], originally developed by Puzio-Kuter and colleagues [11]. This model closely mimics the progression of human disease from NMIBC through MIBC to metastatic stages [16]. Whole-bladder RNA-sequencing analysis revealed significant alterations in the PI3K/AKT/mTOR pathway [16], implicated in 72% of patients with advanced BC [17]. Comparative analysis with established BC subtypes showed a strong similarity to basal-squamous BC [16], prevalent in 35% of patients with MIBC [17], that makes it relevant to use for preclinical investigations. Even though p53 loss of function induces genomic instability [18], one deficiency of this model, as most of GEMM, is its poor mutational burden.

To fill the lack of available relevant models for advanced BC and to easily follow tumor growth and progression, we aimed to generate new models of traceable advanced BC that develop all features of the human disease, including the expression of a known antigen. Our research demonstrates that tumor development in our inducible model mirrors the classical *Tp53/Pten* deletion model without affecting the tumor microenvironment or response to anti-PD-1 therapy. We also developed a novel traceable orthotopic muscle-invasive mouse model of BC that efficiently develops distant metastases, offering a promising and valuable tool for exploring metastatic BC physiopathology and improving immunotherapies.

## 2. Materials and Methods

### 2.1. Ethics

All animal experiments were performed in compliance with the University of Lausanne Institutional regulations and were approved by the veterinarian authorities of the Canton de Vaud [authorizations 3594 (valid from 2020 to 2023) and VD3856 (valid from 2023 to 2026)].

### 2.2. Inducible Mouse Models of Bladder Cancer

The p53^fl/fl^pten^fl/fl^Luc-SIY strain results from a cross between a p53^fl/fl^pten^fl/fl^ mouse [16] and a mouse strain carrying a transgene encoding for a T-cell antigen (SIY) fused with the luciferase (Luc) (B6.129S4-Gt(ROSA)26Sortm3(CAG-luc)Tyj/J, Jackson Laboratories). Male and female mice from 7 to 9 weeks old were used. To induce the deletion of the floxed genes, *tp53* and *pten*, and the floxed STOP-cassette which inhibits the expression of the Luc-SIY fusion protein, a nonreplicative adenovirus that expressed the Cre recombinase under the control of a CMV promoter (Appendix A) was used [#AVL(VB181004-1095pzc)-K1, VectorBuilder, Chicago, IL, USA]. To induce bladder-specific deletion, 2.5 × 10^8^ plaque-forming units of Cre-expressing adenoviral vector in 5 μL of DMEM/hexadimethrine bromide (8 mg/mL) was injected into the bladder lumen of mice by microsurgery, as already described [19]. 

### 2.3. Generation of Luc-SIY-Expressing Bladder Tumor Cell Line

To isolate bladder tumor cells, Luc-SIY-expressing bladder tumors were cut mechanically with scissors and digested for 30 min at 37 °C in medium with 0.1 mg/mL DNase I (#D4527, Sigma, Saint Louis, MO, USA) and 1 mg/mL Colagenase I (#17100017, ThermoFisher Scientific, Waltham, MA, USA). Afterward, tissues were mashed through a 70 μm cell strainer (#352350, Milian, Milan, Italy) in a Petri dish and collected in a 15 mL Falcon tube. With centrifugation for 30 min at 2000 rpm in a density gradient of 75%/100% Percoll (#17-0891-01, GE Healthcare Life Sciences, Marlborough, MA, USA), bladder tumor cells were isolated and washed in 10 mL of RPMIc, before being kept in culture in RPMIc. Luc-SIY-expressing clones were generated by culturing the isolated tumor cells in a limiting dilution assay. An in vitro Luciferase Assay System reagent (#E2610, Promega, Fitchburg, WI, USA) was used to confirm the luciferase activity, according to manufacturer’s instructions. 

### 2.4. Heterotopic Bladder Cancer Mouse Model

Luc-SIY bladder tumor cells were injected subcutaneously (8 × 10^5^ cells in 100 µL of PBS) in immunocompetent mice (C57BL/6 mice, Jackson Laboratories, Bar Harbor, ME, USA), in lymphocyte-deficient mice (Rag1 KO, Jackson Laboratories), or p53^fl/fl^pten^fl/fl^Luc-SIY mice. The tumor’s volume was measured with a caliper. 

### 2.5. Orthotopic Syngeneic Muscle-Invasive Bladder Cancer Mouse Model

For the orthotopic BC mouse model, mice bladders were exteriorized by microsurgery, and urine was removed with disinfected forceps pressure. 1 × 10^6^ cells in 15 µL of DMEM were injected into the bladder lumen using a Hamilton syringe. Bladders were gently put back in the abdomens. Peritonea were sutured using absorbable thread (#V385H, Ethicon), and skin was closed with wound clips (#12022-19, Fine Science, Foster City, CA, USA). 

### 2.6. In Vivo/Ex Vivo Bioluminescence Imaging

Mice were shaved and injected intraperitoneally with 100 µL of 30 mg/mL D-luciferine Firefly (L-8220, Biosynth, Staad, Switzerland). Five to ten minutes after luciferin injection, mice were anesthetized using isoflurane vaporizer and placed inside the camera box of the Xenogen IVIS Lumina II imager (Caliper Life Sciences). Ex vivo metastatic organs were dissected just after the in vivo imaging and placed in a 12-well plate for imaging. The acquisition time was 30 s, with a binning of 4–8 and an f-stop of 1,2. For the bioluminescence signal results from the photon flux emitted by the luciferase-expressing tumor cells, images were analyzed using Living Image© in vivo imaging software 4.7.3 (IVIS Imaging Systems). Regions of interest (ROIs) were drawn around the bladder or metastatic areas, and the number of photons per second within each ROI were recorded. The ROI size was held constant across all images.

### 2.7. Histology and Immunohistochemistry

Bladders were collected from mice at different stages and either fixed in formalin (#HT501128, Sigma) for hematoxylin and eosin staining or embedded in Tissue-Tek® O.C.T.™ Compound (#BCH3979, Sysmex, Kobe, Japan) and frozen for immunohistochemistry (IHC). Tissue sections and hematoxylin-and-eosin staining were performed by the Mouse Pathology Facility (University of Lausanne). For IHC, after a step of fixation with acetone for 30 min on ice and rehydration with PBS for 20 min at room temperature, sections were blocked with PBS containing 0.1% BSA and 1% normal mouse (10410, ThermoFisher) and donkey (#S30, Sigma) serums. Slides were incubated with primary antibodies at 4 °C overnight in PBS containing 0.1% BSA and 1% normal mouse serum. After three washes with PBS, slides were incubated with secondary antibodies for 1 h at room temperature in PBS containing 0.1% BSA and 1% normal mouse serum. The antibodies are listed in Table 1. Finally, sections were washed in PBS and mounted with Fluoromount-GTM with DAPI (#00-4959-52, Invitrogen, Carlsbad, CA, USA). Samples were acquired on Axiovision microscope (Zeiss) with Axiovision v4.8 software, objectives ×10 or ×20. Th images were analyzed using Image J software (http://imagej.nih.gov/ij/, version 1.54g) and QuPath software 0.4.3 [20].

### 2.8. Single-Cell Preparation and FACS Staining

After the digestion step described in Section 2.3 (Generation of Luc-SIY-Expressing Bladder Tumor Cell Line), leucocytes were isolated from bladders in a density gradient of 40%/70% Percoll for 30 min at 2000 rpm. The interphase was collected, and leukocytes were washed in 10 mL of RPMIc. Isolated cells were resuspended in RPMIc and plated for staining. For intracellular cytokines staining, cells were restimulated with anti-CD3 (1 μg/mL, #100314, Biolegend, San Diego, CA, USA) and anti-CD28 (1 μg/mL, #102112, Biolegend) for 6 h at 37 °C in the presence of Golgi Plug (1/1000, #51-2301KZ, BD Biosciences, Franklin Lakes, NJ, USA) before staining. Then, FcγRs were blocked for 15 min at RT with anti-CD16/32 (1/1000, #101320, Biolegend). Cells were incubated with the corresponding extracellular antibodies diluted with FACS buffer for 30 min in the dark at room temperature. Cells were then stained with LIVE/DEADTM Fixable Aqua Dead Cell Stain Kit (#L34966 ThermoFisher Scientific) or Zombie Aqua Fixable Viability kit (#423105 from Biolegend) for 15 min at room temperature in the dark. Cells were fixed with the Foxp3 Transcription Factor Staining Buffer Set (#00-5523-00, eBiosciences, San Diego, CA, USA). Intracellular staining was performed in permeabilizing buffer (#00-8333-56, Invitrogen) and incubated for 1 h at room temperature. The antibodies used for the flow cytometry are listed in Table 2. Data acquisition was performed on a FORTESSA machine. FlowJo software 10.10.0 was used to analyze the FACS data.

### 2.9. Therapeutic Treatments

Treatments were started 8 weeks after tumor initiation, when tumors reached the muscle-invasive stage and were palpable. Four injections of PD-1 blocking Ab (300 μg/dose, RMP1-14 clone, BioXcell, Lebanon, NH, USA) or IsoCT (300 μg/dose, 2A3 clone, BioXcell) were administered by intraperitoneal injections every 2–3 day for 8 days. Nine days after the beginning of the treatments, mice were sacrificed for the analysis of the immune microenvironment or kept for a survival experiment.

### 2.10. Statistical Analysis

GraphPad Prism 10 software was used for graph generation and biostatistical analyses. One-way ANOVA and two-way ANOVA were first used for multiple group comparison and time follow-up, respectively. Then, specific statistical tests used for each specific experiment are detailed in the legend of the corresponding figure. For nonsignificant differences, *p*-values are absent, while for statistically significant results, the following *p*-values are included in the figures: * *p* < 0.05; ** *p* < 0.01; *** *p* < 0.001; **** *p* < 0.0001.

## 3. Results

### 3.1. Luc-SIY Transgene Does Not Influence Bladder Cancer Development

Our team previously characterized a genetically engineered mouse model of BC that mimics BC development in patients and is resistant to anti-PD-1 treatment [16]. To monitor tumor progression, as well as metastasis in vivo and, at the same time, introduce a known T-cell antigen into our model, we crossed p53^fl/fl^pten^fl/fl^ mice with stop^fl/fl^Luc-SIY mice, further called p53^fl/fl^pten^fl/fl^Luc-SIY mice. The injection of an adenoviral vector encoding for the Cre-recombinase efficiently induced bladder tumor development in both p53^fl/fl^pten^fl/fl^Luc-SIY^−^ and p53^fl/fl^pten^fl/fl^Luc-SIY^+^ mice, hereafter referred to as the “inducible” BC models. These tumors transitioned from NMIBC to MIBC and advanced MIBC stages at specific time points following adenovirus injection: 5 weeks (5 w), 9 weeks (9 w), and 12 weeks (12 w), respectively. The hematoxylin and eosin (H&E) staining showed concurrent tumor development in both genetic strains (Figure 1A). At 5 w, tumors were confined to the superficial layers of the bladder (i.e., urothelium and lamina propria). By 9 w, all tumors had invaded the muscle layers. By 12 w, the bladder was entirely infiltrated by tumor cells, obscuring the different structures and exhibiting a metastatic rate of 11.76% (2/17). Tumor mass increased over time and remained comparable between genotypes at each time point (Figure 1B). The expression of the Luc-SIY transgene, controlled by a Flox-STOP-Flox cassette, was activated upon Cre-recombinase delivery. Bioluminescent imaging revealed exclusive luciferase activity in the Luc-SIY-expressing tumors (Figure 1C). These findings confirm that the p53^fl/fl^pten^fl/fl^Luc-SIY^+^ mouse strain developed bladder tumors that expressed the Luc-SIY transgene. Upon comparison of the Luc-SIY^−^ and Luc-SIY^+^ tumors, we observed similar patterns of muscle-invasive development. 

### 3.2. Luc-SIY Transgene Has No Influence on the Tumor Immune Microenvironment Composition

Even if no difference in tumor growth was noticed, the presence of the Luc-SIY transgene could create a more immunogenic tumor compared to our initial model [16]. To explore the impact of the Luc-SIY transgene expression on the immune tumor microenvironment (TME), we analyzed the immune cell composition over time and across genotypes. In both Luc-SIY^−^ and Luc-SIY^+^ tumors, CD4^+^ and CD8^+^ T cells were localized within the tumor core (Figure 2A), with no difference in CD4^+^ and CD8^+^ T-cell numbers (Figure 2B,C). CD25, the interleukine-2 receptor alpha, was utilized to discriminate conventional CD4 T cells (CD25^−^) and T-regulatory cells (CD25^+^) among the CD4^+^ population. Approximately 80% to 90% of the CD4^+^ T cells were conventional CD4 T cells (CD25^−^), while T-regulatory cells (CD25^+^) accounted for 10 to 20%, without a difference between the groups (Figure 2D). Expression of CD44 and PD-1 on CD8^+^ T cells did not differ across groups (Figure 2E,F). Regarding their functionality, both CD8^+^ and CD4^+^ T cells showed no change in their capacity to produce IFNγ or TNFα upon restimulation (Figure 2G–J). An analysis of the various myeloid cells also did not reveal differences between the groups (Figure 2K,L). Taken together, these findings showed that expression of the Luc-SIY transgene did not modify the immune cell composition and activation state of the TME in our inducible model of BC. 

### 3.3. Luc-SIY^+^ Inducible Bladder Cancer Model Is Resistant to Anti-PD-1 Immune Checkpoint Blockade

Despite the absence of endogenous T-cell reactivity toward Luc-SIY-expressing bladder tumors, we investigated whether CD8 T cells could respond more efficiently to immune checkpoint blockade in Luc-SIY^+^ tumor-bearing mice. To this end, four injections of either anti-PD-1 or isotype control (IsoCT) antibodies were performed i.p. at 8 weeks post-tumor induction every 2–3 days in Luc-SIY^+^ tumor-bearing mice. Monitoring of the luciferase activity post-treatment revealed no significant difference between the anti-PD-1- and IsoCT-treated groups (Figure 3A). Similarly, no difference was observed in survival rates between the two groups (Figure 3B). In parallel, the composition of the TME of Luc-SIY^+^ tumors was assessed 9 days following the beginning of treatment. Examination of the T-cell compartment revealed no significant change in CD8^+^ cell numbers within the tumor (Figure 3C). The proportion of CD44^high^, IFNγ^+^, and TFNα^+^ CD8^+^ T cells did not significantly increase between the groups (Figure 3D–F). Similar results were observed for CD4^+^ T cells (Figure 3G–J). Altogether, these findings demonstrate that this model is unresponsive to anti-PD-1 treatment, despite expression of the Luc-SIY transgene.

### 3.4. Luc-SIY-Expressing Bladder Tumors Are Immunogenic in C57BL/6 Mice

To evaluate the efficacy of Luc-SIY protein expression for immune recognition and the potential induction of an antitumor immune response, we subcutaneously injected Luc-SIY^+^ and Luc-SIY^−^ cell lines isolated from our inducible BC model into C57BL/6 mice. The slower growth rate of Luc-SIY^+^ tumors, as compared to Luc-SIY^−^ tumors, indicates a probable antigen-mediated effect on tumor progression (Figure 4A). To test the role of the adaptative immune system in this tumor control, we injected Luc-SIY^+^ tumors into Rag1 knockout (KO) mice and C57BL/6 mice (Figure 4B). Tumor growth was faster in Rag1 KO mice, suggesting T- or B-cell involvement in the control of tumor growth. As tumor growth was not impacted in the inducible model despite the presence of the Luc-SIY transgene, we hypothesized that tolerance mechanisms were preventing tumor rejection in p53^fl/fl^pten^fl/fl^Luc-SIY mice. Central and peripheral tolerances have been shown to play a significant role in a similar Luc-SIY Cre-LoxP system of a lung adenocarcinoma [21]. To test this, we subcutaneously injected Luc-SIY-expressing BC cell lines into C57BL/6 or p53^fl/fl^pten^fl/fl^Luc-SIY mice. Tumors developed more slowly in C57BL/6 mice than in p53^fl/fl^pten^fl/fl^Luc-SIY mice, confirming this hypothesis (Figure 4C). Considering that heterotopic tumors do not accurately recapitulate the TME from the bladder, we repeated these experiments by injecting the isolated Luc-SIY-expressing BC cell line into the bladders of both C57BL/6 mice and p53^fl/fl^pten^fl/fl^Luc-SIY mice. In the C57BL/6 mice, a loss of the bioluminescence signal was observed 13 days post-cell injection (Figure 4D), whereas in the p53^fl/flp^ten^fl/fl^Luc-SIY mice, the signal progressively increased (Figure 4E). The P53^fl/fl^pten^fl/fl^Luc-SIY mice were sacrificed at day 43, while C57BL/6 mice were sacrificed at day 84, exhibiting the same mean bladder weight (Figure 4F). This suggests that Luc-SIY tumor cells were initially controlled in C57BL/6 mice, but after the loss of expression of the Luc-SIY transgene, tumors finally developed. Altogether, Luc-SIY tumors are immunogenic, but tolerance mechanisms prevent tumor control in our inducible model.

### 3.5. Orthotopic Tumor Cell Injection as a Novel Syngeneic Mouse Model of Metastatic MIBC

To test whether Luc-SIY BC cell lines can be used as a metastatic model of BC, Luc-SIY-expressing tumor cells were injected into the bladder of p53^fl/fl^pten^fl/fl^Luc-SIY mice. Histopathological analysis using H&E staining confirmed the development of muscle-invasive tumors in this model (Figure 5A). Bioluminescence imaging confirms the maintenance of the luciferase activity over tumor development (Figure 5B). Moreover, an increasing bioluminescence signal was measured at distant sites from the bladder (5/6, no metastases in mouse 1) in vivo (Figure 5B). Ex vivo bioluminescence analyses of organs confirmed the 100% incidence of bladder tumor, as well as metastases formation in the spleen (3/6), in the draining lymph node (DLN) (1/6), and in the pancreas (5/6) (Figure 5C,D). In this model, only 16.67% of the mice (1/6) did not develop any metastasis, 66.66% (4/6) developed one or two metastases, and 16.67% (1/6) developed three metastases (Figure 5E), illustrating the aggressiveness and reproducibility of this model. Altogether, these findings indicate that this orthotopic syngeneic mouse model of BC successfully generates invasive tumors that can spread toward distant organs and can be monitored in live animals.

## 4. Discussion

BC is a smoking-related cancer that is associated with a high mutational burden [4] and displays an unfavorable prognosis with few therapeutic options in its advanced stages. The efficacy of immunotherapies relies on the immune system’s ability to identify and engage a powerful immune response against TAAs [22]. These molecular signatures, aberrant or mutated proteins expressed by cancer cells, offer numerous targets for eliciting a specific antitumor response. Adequate and reproducible in vivo models recapitulating the human disease are indispensable to understand the physiopathology of BC but also to develop and test new therapeutic strategies. A significant limitation in preclinical research is the lack of mouse model of BC that faithfully mimics the intricate TAA profile observed in human pathology. 

Currently, the most relevant model that recapitulates the high mutational burden of BC is the BBN model, which still presents important differences in terms of mutations with the human pathology [9]. Apart from the heterogeneity of tumor development, the BBN model necessitates bladder palpation or urine inspection, reflecting a late tumor stage, before injecting treatments that render it difficult to use in preclinical settings [10]. For this reason, in vivo imaging systems are useful in monitoring tumor development at an early stage and allow for a precise evaluation of therapeutic responses. Noninvasive in vivo imaging also limits the number of animals used.

To mimic TAA expression of human BC and follow tumor development in vivo in a reproducible model of BC, we crossed a p53^fl/fl^pten^fl/fl^ mouse strain with mice expressing a construct that fuses the SIY antigen with luciferase. We demonstrated in this study that the luciferase expression properly allowed for the follow up of tumor development. At the same time, the transgene enables the expression of a TAA in bladder tumors. We showed that, in our inducible model of BC, the expression of the Luc-SIY transgene did not influence tumor development toward the muscle-invasive stage or the TME composition. Moreover, the presence of the Luc-SIY transgene did not induce response to anti-PD-1 treatment, as already described in our initial inducible model [16]. This model is representative of the majority of patients with BC with the presence of TAAs and by its resistance to anti-PD-1 therapy, which make it relevant to decipher the mechanism of ICI resistance and test new therapeutic strategies. This last observation aligns with the findings of Cheung and coworkers, who used the same Luc-SIY Cre-LoxP system in a lung adenocarcinoma mouse model. They demonstrated that antigen-specific T cells had to overcome tolerance to initiate an effective antitumor response [21]. In the present study, we also confirmed the tolerance of our inducible model by injecting Luc-SIY-expressing tumor cells in immunocompetent C57BL/6 mice and immunocompromised mice. 

Following the cell injection into C57BL/6 mice, we also observed a loss of bioluminescent signal during the second week, while the tumor continued to grow, as reported after the injection Luc-expressing MB49 cells [6]. It is likely that within C57BL/6 mice, the TME triggers a process of clonal selection among tumor cells, a mechanism part of the “immune evasion” concept [23]. The immune system can mount a response against the immunogenic cell clones expressing the transgene, whereas other clones, which may have lost it, potentially evade immune detection, and proliferate. This mechanism also explains why tumors grew more slowly in the C57BL/6 mice compared to the p53^fl/fl^pten^fl/fl^Luc-SIY mice.

Also lacking in the field of BC is the availability of metastatic models to decipher the biological processes that promote metastases [24]. Several GEMM targeting P53, PTEN, RB1 or NOTCH genes have described the progression from NMIBC toward MIBC, but the development of metastases is relatively low in these models [7,11,25,26]. In this study, we developed a syngeneic model of metastatic BC by injecting Luc-SIY-expressing cells extracted from our inducible MIBC model into the bladder lumen. The tumor incidence rate in the bladder and metastatic rate were 100% and 83%, respectively. However, metastases were mainly localized in the pancreas and the spleen, which is not a main pattern of metastasis spread in human BC [27]. Cai and coworkers developed a metastatic model of BC that invades organs involved in human diseases, such as ureter, lung, and liver. This model is based on the injection of the MB49 cell line in the bladder wall [28]. The disadvantages of this model compared to ours are rapid tumor growth and mice death, which reduce the therapeutic window for preclinical investigations, in addition to the fact that the tumor did not arise from the urothelium [28]. Recently, a quadruple knockout (Rbl1^−/−^; Rb1^F/F^; TrP53^F/F^; Pten^F/F^) mouse model was described to develop advanced BC with a metastatic development rate of 100% in the liver and lungs [12,14]. These tumors are genetically close to the human disease, which makes it relevant for preclinical settings, but it does not offer the possibility to perform in vivo tracing of metastasis spread and development, which requires necropsy [14].

Even if syngeneic models allow for the development of tumors in immunocompetent mice and are essential for the evaluation of immunotherapies [29], injecting a large number of tumor cells is very different from the natural evolution of a cancer, inducing some inflammation, limiting the clonal evolution of tumor cells and, by consequence, the intratumor heterogeneity. Our model of metastatic BC does not recapitulate the large TMB and the genomic heterogeneity found across patients with BC. Otherwise, this syngeneic model of metastatic BC offers several advantages compared to the models already available; the incorporation of luciferase allows for the detection of distant metastasis before necropsy; and this model is distinguished by its rapidity, simplicity of execution, and reproducibility, making it an invaluable resource for therapeutic studies. It also offers the possibility of performing genetic manipulations in tumor cells to understand BC’s complex biology during its evolution, response to treatment, or metastatic process.

## 5. Conclusions

In conclusion, this study introduced novels inducible and syngeneic orthotopic mouse models of advanced MIBC, offering a new tool to study BC. Our findings demonstrate the model’s utility in exploring tumor initiation, elucidating the immune system’s involvement, and evaluating responses to ICIs. Importantly, the model’s ability to replicate tumor invasion through muscle layers and its potential to develop distant metastasis make it a valuable tool for furthering our understanding of both fundamental and translational aspects of BC.

## Figures and Tables

**Figure 1 cancers-16-02245-f001:**
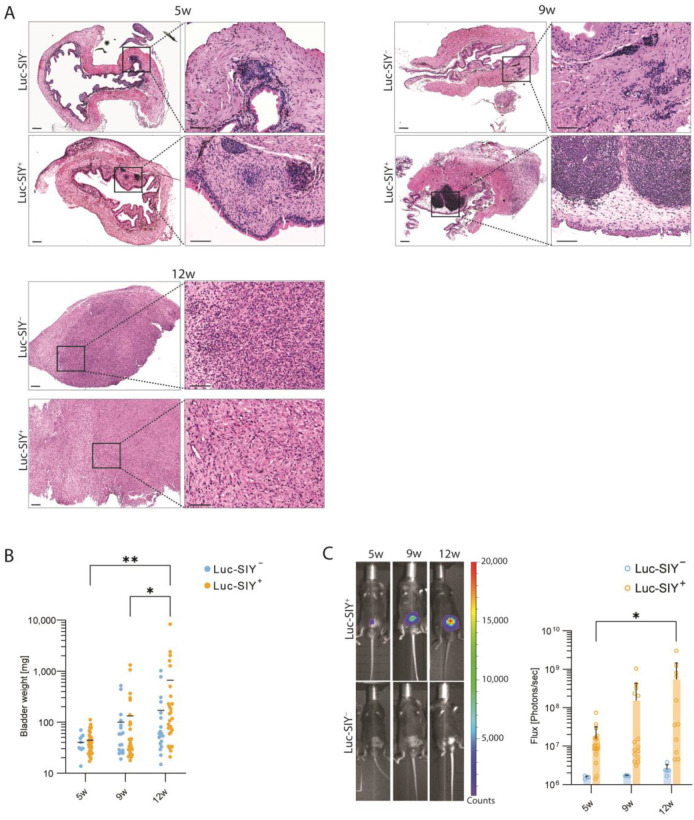
Tumor progression in Luc-SIY^−^ and Luc-SIY^+^ mice. (**A**) Representative histological images showing bladder tumor development in Luc-SIY^−^ and Luc-SIY^+^ mice at 5 weeks (5 w), 9 weeks (9 w), and 12 weeks (12 w) post-tumor induction. Scale bars are 200 µm for low magnification and 100 µm for high magnification. (**B**) Bladder weight of the Luc-SIY^−^ and Luc-SIY^+^ tumor bladders at 5 w, 9 w, and 12 w post-tumor induction. Each dot represents an individual mouse, and bars represent the mean. Two-way ANOVA, followed by the Tukey test. (**C**) Representative in vivo bioluminescent images and quantification of the flux (photons per second) of Luc SIY^−^ and Luc-SIY^+^ mice at 5 w, 9 w, and 12 w post-tumor induction. Images were acquired 5 min after luciferin injection with a binning of 4. * *p* < 0.05; ** *p* < 0.01.

**Figure 2 cancers-16-02245-f002:**
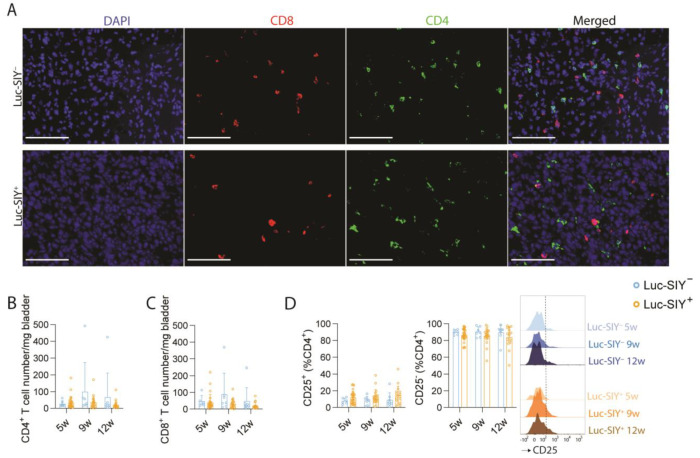
Immune profiling of Luc-SIY^−^ and Luc-SIY^+^ bladder tumors. (**A**) Representative DAPI (blue), CD8 (red), CD4 (green), and merged immunofluorescence images of Luc-SIY^−^ and Luc-SIY^+^ bladder tumors at 12 w post-tumor induction. Scale bars = 100 µm. (**B**,**C**) Representative bar plots of Luc-SIY^−^ and Luc-SIY^+^ bladder tumors at 5 w, 9 w, and 12 w post-tumor induction of the number per mg of bladder of CD4^+^ T cells (**B**) and CD8^+^ T cells (**C**). (**D**–**F**) Representative FACS histograms and bar plots of mean percentage of CD25^+^ and CD25^−^ populations among CD4^+^ T cells (**D**), CD44^high^ population among CD8^+^ T cells (**E**), and PD-1^+^ population among CD44^high^ CD8^+^ T cells (**F**) in Luc-SIY^−^ and Luc-SIY^+^ bladder tumors at 5 w, 9 w, and 12 w post-tumor induction. (**G**–**J**) Representative FACS histograms and bar plots after restimulation of the mean percentage of the IFNγ^+^ population among CD44^high^ CD8^+^ T cells (**G**), TNFα^+^ population among CD44^high^ CD8^+^ T cells (**H**), IFNγ^+^ population among CD44^high^ CD4^+^ T cells (**I**), and TNFα^+^ population among CD44^high^ CD4^+^ T cells (**J**) in Luc-SIY^−^ and Luc-SIY^+^ bladder tumors at 5 w, 9 w, and 12 w post-tumor induction. (**K**) Mean proportions of MHCII^low^ tumor-associated macrophages (TAMs), MHCII^high^ TAMs, immature macrophages, Ly6C^low^ monocytes, Ly6C^int^ monocytes, Ly6C^high^ monocytes, neutrophils, and other cells among CD11b^+^ cells during tumor development in Luc-SIY^−^ and Luc-SIY^+^ bladders, n = 7–28. (**L**) Representative curve of the MHCII^high^/MHCII^low^ TAMs ratio in Luc-SIY^−^ and Luc-SIY^+^ tumor bladders over time. In the bar plots, each dot represents an individual mouse, and the top of the box represents the mean with the SD. Two-way ANOVA, followed by the Tukey test.

**Figure 3 cancers-16-02245-f003:**
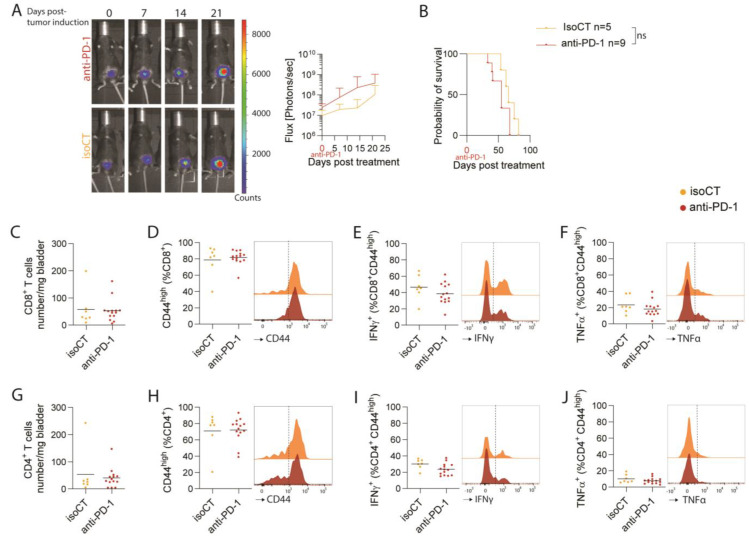
Analysis of anti-PD-1 response in the inducible Luc-SIY^+^ MIBC mouse model. (**A**) Representative in vivo bioluminescent images and quantification of the flux (photons per second) of anti-PD-1- or IsoCT-treated Luc-SIY^+^ tumor-bearing mice. Images were acquired 5 min after luciferin injection with a binning of 4. The mean ± SD is shown; *t*-tests followed by Mann–Whitney tests were used. (**B**) Kaplan–Meier curves of anti-PD-1- or IsoCT-treated Luc-SIY^+^ tumor-bearing mice. Log-rank test were used; NS = nonsignificant. (**C**–**J**) Representative dot plots and FACS histograms after restimulation of a number of CD8+ T cells per mg of bladder (**C**), mean percentage of CD44^high^ population among CD8^+^ T cells (**D**), mean percentage IFNγ^+^ population among CD44^high^ CD8^+^ T cells (**E**), mean percentage TNFα^+^ population among CD44^high^ CD8^+^ T cells (**F**), number of CD4^+^ T cells per mg of bladder (**G**), mean percentage of CD44^high^ population among CD4^+^ T cells (**H**), mean percentage IFNγ^+^ population among CD44^high^ CD4^+^ T cells (**I**), and mean percentage TNFα^+^ population among CD44^high^ CD4^+^ T cells (**J**) of anti-PD-1- or IsoCT-treated Luc-SIY^+^ bladder tumor-bearing mice. Each dot represents an individual mouse and bars represent the mean; *t*-test analysis.

**Figure 4 cancers-16-02245-f004:**
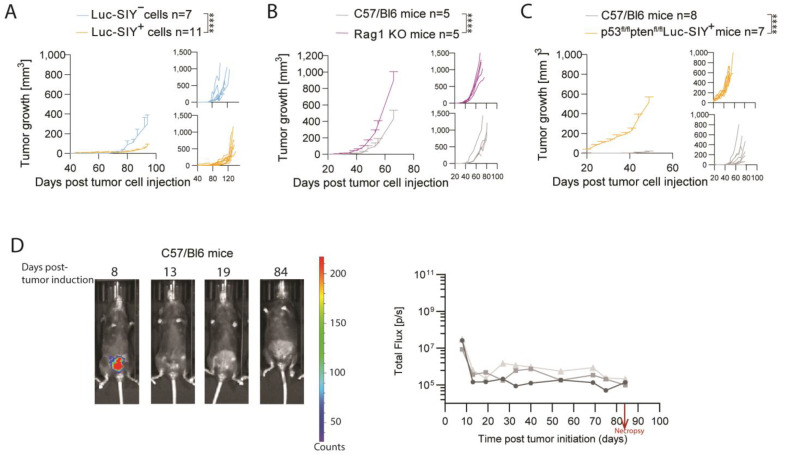
Nonautochthonous injections of derived-bladder tumor cells. (**A**–**C**) Tumor growth of Luc-SIY^+^ and Luc-SIY^−^ tumor cells implanted subcutaneously in C57BL/6 mice (**A**), Luc-SIY^+^ tumor cells implanted subcutaneously in C57BL/6 or Rag 1 KO mice (**B**), and Luc-SIY^+^ tumor cells implanted subcutaneously in C57BL/6 or p53^fl/fl^pten^fl/fl^Luc-SIY mice (**C**). Large graphs represent the mean ± SEM, and small graphs represent an individual mouse in each group. Two-way ANOVA followed by Sidak tests were used. (**D**,**E**) Representative in vivo bioluminescent images and quantification of the flux (photons per second) after orthotopic injection of Luc-SIY-expressing bladder tumor cells in C57BL/6 mice (**D**) and p53^fl/fl^pten^fl/fl^Luc-SIY mice over time (**E**). Each curve represents a separate mouse. Images were acquired 10 min after luciferin injection with a binning of 4. (**F**) Bladder weight of healthy bladders (CT) or bladders 84 days post-tumor cell injection in C57BL/6 mice or 43 days post-tumor cell injection in the p53^fl/fl^pten^fl/fl^Luc-SIY mice. Each dot represents an individual mouse, and bars represent the mean. One-way ANOVA, followed by the Kruskal–Wallis test, was used. * *p* < 0.05; ** *p* < 0.01; **** *p* < 0.0001.

**Figure 5 cancers-16-02245-f005:**
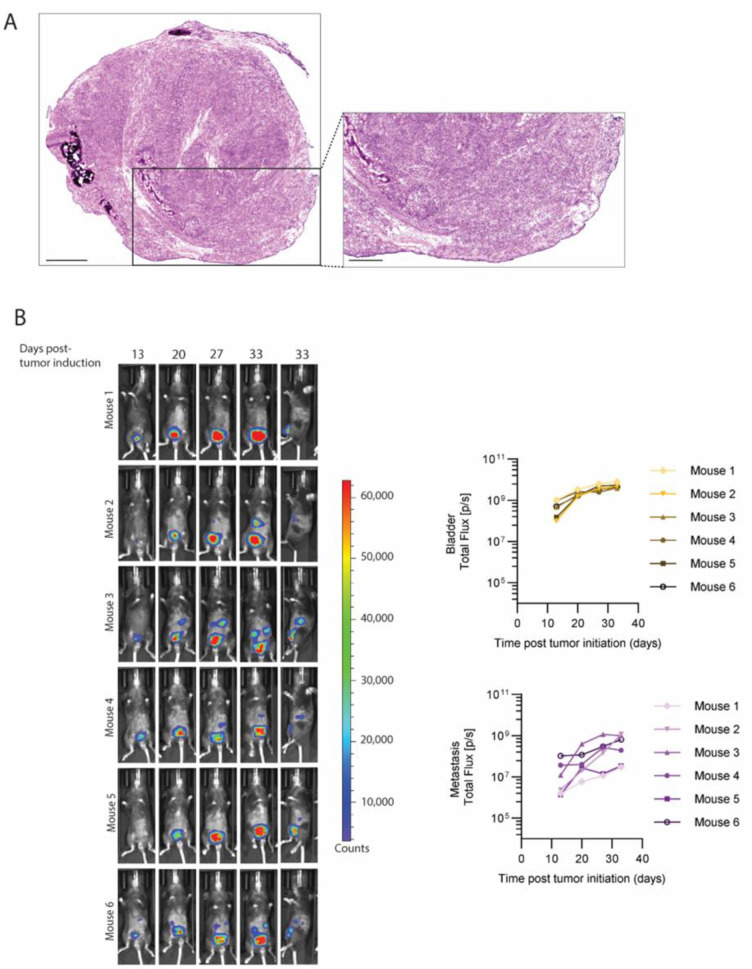
Orthotopic injections of Luc-SIY^+^ bladder tumor cells. (**A**) Representative histological image of orthotopic tumor 7 weeks post-tumor implantation in p53^fl/fl^pten^fl/fl^Luc-SIY mice. Scale bars = 1 mm for low magnification and 500 µm for high magnification. (**B**) Representative in vivo bioluminescent analysis of the bladder and distant signal after orthotopic injection of Luc-SIY-expressing bladder tumor cells in p53^fl/fl^pten^fl/fl^Luc-SIY mouse bladders. Mice were positioned supine for the first four columns of the image and laterally for the last column. Relative quantification of the flux (photons per second) from the bladder and distant signal was determined on the basis of the mice positioned supine. Each curve represents a separate mouse. Images were acquired 10 min after luciferin injection with a binning of 4. (**C**) Ex vivo bioluminescent quantification of bladder, spleen, pancreas, and DLN. The threshold was set regarding the basal signal intensity of each organ. Each dot represents an individual mouse, and bars represent the mean. (**D**) Percentage of metastases in the spleen, pancreas, and DLN. (**E**) Percentage of mice developing none, one, two, or three metastases.

**Table 1 cancers-16-02245-t001:** Antibodies used for immunohistochemistry.

Target	Clone	Reference	Provider	Dilution
CD4	H129.19	550278	BD pharmingen (Franklin Lakes NJ, USA)	1/100
CD8	EPR21769	ab217344	Abcam (Cambridge, UK)	1/100
Anti-rat		712-545-150	Jackson ImmunoResearch (West Grove, PA, USA)	1/500
Anti-rabbit		A32733	Invitrogen (Carlsbad, CA, USA)	1/500

**Table 2 cancers-16-02245-t002:** Antibodies used for flow cytometry.

Extracellular Target	Fluorochrome	Clone	Reference	Provider	Dilution
CD45.2	BV711	104	109847	Biolegend	1/50
CD45.2	BV650	104	109836	Biolegend	1/100
CD4	FITC	GK1.5		FACS Facility UNIL	1/1000
CD4	BV605	RM4-5		FACS Facility UNIL	1/200
CD8	BV786	53-6.7	100750	Biolegend	1/200
CD8	APC efluor 780	53.6.7	100714	Biolegend	1/100
CD11b	BV605	M1/70	101257	Biolegend	1/500
F4/80	PEDz594	BM8	123146	Biolegend	1/50
Ly6C	Percp/Cy5.5	HK1.4	45-5932-82	Invitrogen	1/200
Ly6G	BV650	1A8	127641	Biolegend	1/200
MHCII	BV786	M5/114.15.2	107645	Biolegend	1/1000
CD11c	FITC	N418	11-0114-85	Invitrogen	1/100
B220	PE/Cy7	RA3-6B2	103222	Biolegend	1/1000
B220	BV570	RA3-6B2	103237	Biolegend	1/50
PD-1	BV711	29F.1A12	135231	Biolegend	1/200
CD44	APC/Cy7	IM7	47-0441-82	Invitrogen	1/100
CD25	PE/Cy7	PC61	25-0251-82	eBioscience	1/100
TNFα	BV421	MP6-XT22	506318	Biolegend	1/100
IFNγ	PE/Cy7	XMG1.2	505826	Biolegend	1/100

## Data Availability

All raw data are available upon request.

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
