# Peer review of "Development of Traceable Mouse Models of Advanced and Metastatic Bladder Cancer"

_cancers, 2024, doi:10.3390/cancers16122245_

Round 1
Reviewer 1 Report
Comments and Suggestions for Authors
good work with some details to modified.

Author Response
Reviewer 1
To the authors of the manuscript: “Development of traceable mouse model for advanced bladder cancer”.
I think that the authors have approached the work in a direct and precise manner. All the sections, materials and methods and legends, are well described and defined. It addresses one of the main issues in these field, the generation of animal models of advanced and/or metastatic disease for the study of the behaviour tumor and therapies. I think it is a well thought out and executed work.
We thank the reviewer for these positive comments.
Even so, I would like to have some things reflected in the manuscript or clarified to me, which I will explain below:
- In Figure 1, since it deals with the description of the model, they should include the 12-week images as well. I think it would be more correct to present a graph with Bioluminescent signal quantification (photons/s/cm2/sr) of regions of interest was carried out using Living Image software, for example, or try to measure the volume (3D), better than bladder size. This is also the case in Figure 4F.
We thank the reviewer for the suggestions. We added the 12-weeks H&E images to the figure 1A and the bioluminescence signal quantification of the figure 1C. For the figure 4F, we cannot provide tumor volume for these results. However, the bladder weight reflects the size/stage of the bladder as we already showed in our previous publication (Leblond et al., CIR, 2020).
- Line 208 : the authors administer the treatment after 8 weeks from tumor onset and before the beginning of the invasive stage. Each animal is different. How could the authors have determined this “timepoint” of when the invasive stage started?
We apologize for the lack of clarity. In our settings, we have shown that the transition from NMIBC to MIBC occurs between 6 weeks and 8 weeks post-tumor induction for all the animals. We start the treatment at 8 weeks because we can palpate by hand the bladder and feel whether the tumor is already invasive or not. When tumors reach the muscle, the bladder becomes harder at palpation. This allows us to make the group homogeneous before treatment. We clarified this in the text (lines 213-214):
“Treatments were started 8 weeks after tumor initiation, when tumors reached the muscle-invasive stage and were palpable.”
- In the discussion they comment on the shortage of models in both advanced and metastatic disease. But they do not comment on one of the invasive metastatic models that resembles human disease really well (Rbl1-/- ; Rb1 F/F; TrP53F/F; PtenF/F). Please, since there are few models, it is better to include them. In this model, for example the same technique of lumen instillation is used but with an adenovirus under the control of the keratin 5 promoter, urothelial (basal) cell specificity. Much more targeted. Puzio's paper, has some points on which I disagree. For example, they use adenoviruses with the CMV promoter which is ubiquitous and many of their tumors remind me of mesotheliomas.
We thank the reviewer to highlight this gap in our manuscript. We corrected that and we included it in
-the introduction (lines 82-87):
“One main difficulty to generate these models is the limited number of promoters displaying bladder-specific expression [5]. To overcome this problem, the use of adenoviral vector encoding for the Cre-recombinase directly delivered into the bladder of mice with floxed tumor suppressor genes has been developed.”
- and in the discussion (lines 457-462):
“Recently, a quadruple knockout (Rbl1-/-;Rb1F/F;TrP53F/F;PtenF/F) mouse model was described to develop advanced BC with a rate of metastatic development of 100% in the liver and the lung. The tumors are genetically close to the human disease that makes it relevant for preclinical settings but it does not offer the possibility to perform in vivo tracing of metastasis spread and development, which requires necropsy.”
In addition, tumor development goes from 0 to 6 months and does not agree with the timepoints of this work. The females present tumors after 6 months and also a 75% probability of metastasis... Can you explain this?
We agree with the reviewer that in our hand, tumor development is faster than what was described in the paper of Puzio-Kuter et al, 2009. In this paper, at 6 months, all mice were dead. The median survival was about 4 months. In ours, all mice die around 3 months as already described in our previous study. We do not have a clear explanation for the differences between the two studies beside the difference in animal facilities and microbiota, and possibly the Adenovector providers. One common feature between the 2 studies is the absence of difference for tumor growth between males and females.
- The authors comment on the adenovirus used in their assays (AVL(VB181004-1095pzc)-K1): can you describe a little of what this adenovirus consists of?
We apologize for the lack of clarity. We added a figure (Supplemental Figure 1) and detailed the adenovirus used (lines 116-123):
“To induce the deletion of the floxed genes, tp53 and pten, and the floxed STOP-cassette which inhibit the expression of the Luc-SIY fusion protein (Supplemental Figure 1), a non-replicative adenovirus that expressed the Cre recombinase under the control of a CMV promoter is used [#AVL(VB181004-1095pzc)-K1, VectorBuilder, Chicago, IL, USA]. To induce bladder-specific deletion, 2.5x108 plaque-forming units of Cre-expressing adenoviral vector in 5μl of DMEM/hexadimethrine bromide (8mg/ml) were injected into the bladder lumen of mice by micro-surgery as already described [19].”
Off note, despite the presence of a CMV promoter, our previous transcriptomic and histologic analysis of the tumors showed that they are comparable to MIBC.
- Can the authors give any explanation for the appearance of metastatic foci in unusual places? No liver or lung metastases have been found. Please include a possible explanation.
We understand the reviewer’s concern but have so far no explanation for the unusual place of metastatic formation. Some studies have shown that BC could metastasize in pancreas (DOI:10.7759/cureus.14851) or spleen (DOI: 10.2214/AJR.10.5036), but in rare cases. We are currently investigating the involvement of different pathways in the process of metastatic formation in our model to try to explain why the metastasis in our model preferentially developed in these organs.
Finally, I would like the authors to remind that syngeneic models allow the tumor to develop in an intact microenvironment and with an active immune system, but they have some important disadvantages when exploring the results, such as the alteration of the non-human host stroma, the variable clinical predictive value and the questionable relationship with naturally occurring human disease.
We discussed in more details the advantages and disadvantages of our syngeneic model (lines 463-474):
“Even if syngeneic models allow the development of tumors in immunocompetent mice and are essential for the evaluation of immunotherapies, injecting a large number of tumor cells is very different from the natural evolution of a cancer, inducing some inflammation, limiting the clonal evolution of tumor cells and by consequence the intratumor heterogeneity. Our model of metastatic BC does not recapitulate the large TMB and the genomic heterogeneity found across BC patients. Otherwise, this syngeneic model of metastatic BC offers several advantages compared to the models already available: the incorporation of the luciferase allows the detection of distant metastasis before necropsy; and this model is distinguished by its rapidity, simplicity of execution, and reproducibility, making it an invaluable resource for therapeutic studies. It also offers the possibility to perform genetic manipulations in tumor cells to understand BC's complex biology during its evolution, response to treatment or metastatic process.”
Reviewer 2 Report
Comments and Suggestions for Authors
A very interesting model. This study introduced novels inducible and syngeneic orthotopic mouse models of advanced muscle invasive bladder caner, offering a new tool to study BC. Here are some questions:
- In figure 1, can you also show the HE staining in 12 weeks, and could you compare some cancer development markers by IHC in LUC-SIY- and LUC-SIY+ tissues?
- In figure 1B, it is clear that the tumor mass increased over time, but the difference of tumor mass between genotypes is not clear, especially in 12 w, please include the P value between genotypes.
- The results part should be describe more reasonable, like the Figure 2D, What does CD4+ with CD25- or CD25+ mean?
- The cells injected into C57/Bl6 mice, and it is observed a loss of biolumines-421 cent signal during the second week. Can you track tumor development in models other than the p53fl/flptenfl/flLuc-SIY mouse?
Author Response
A very interesting model. This study introduced novels inducible and syngeneic orthotopic mouse models of advanced muscle invasive bladder cancer, offering a new tool to study BC. Here are some questions:
We thank the reviewer for these comments.
- In figure 1, can you also show the HE staining in 12 weeks, and could you compare some cancer development markers by IHC in LUC-SIY- and LUC-SIY+ tissues?
We thank the reviewer for the suggestions. We added the 12-weeks H&E images to the figure 1A. For the cancer development markers, we observed by H&E that the tumor progress through the different layers similarly between the two genotypes. Unfortunately, the short time allowed for revision does not allow us to perform additional experiments to characterize markers of evolution during the progression of the tumor.
- In figure 1B, it is clear that the tumor mass increased over time, but the difference of tumor mass between genotypes is not clear, especially in 12 w, please include the P value between genotypes.
The p value at 12w between the two genotypes is p=0.18. Only p values with p<0.05 are indicated on the graph, as indicated in the legend of figure B.
- The results part should be describe more reasonable, like the Figure 2D, What does CD4+ with CD25- or CD25+ mean?
We apologize for the lack of clarity. We detailed the use of CD25 in the result part (lines 267-270):
“CD25, the interleukine-2 receptor alpha, was utilized to discriminate conventional CD4 T cells (CD25-) and T regulatory cells (CD25+) among the CD4+ population. 80% to 90% of CD4+ T cells were conventional CD4 T cells (CD25-) while T regulatory cells (CD25+) accounted for 10 to 20%, without difference between groups (Figure 2.D).”
- The cells injected into C57/Bl6 mice, and it is observed a loss of bioluminescent signal during the second week. Can you track tumor development in models other than the p53fl/flptenfl/flLuc-SIY mouse?
The immune response against the Luc-SIY pushes the selection of tumors cells without the expression of the transgene. In C57Bl/6 mice, as mentioned by the reviewer, we lose the expression of the luciferase quite rapidly and cannot follow tumor growth using this method. For the C57Bl/6, in the heterotopic model, subcutaneous tumors were measured with a caliper that allows us to follow tumor growth. For the orthotopic model, we could only follow the tumor growth by palpation and by weighing the bladder at the end of the experiment.
Reviewer 3 Report
Comments and Suggestions for Authors
Dear author
The manuscript needs the map and definition of the vector "The p53fl/flptenfl/flLuc-SIY strain results from a crossing between a p53fl/flptenfl/fl mouse 106 [9] with a mice carrying a transgene encoding for a T cell antigen (SIY) fused with the 107 luciferase (Luc) (B6.129S4-Gt(ROSA)26Sortm3(CAG-luc)Tyj/J, Jackson Laboratories). To 108 induce the deletion of p53 and pten genes and induce the expression of the Luc-SIY fused 109 protein into the bladder, 2.5x108 plaque-forming units of Cre-expressing adenoviral vector 110 [#AVL(VB181004-1095pzc)-K1, VectorBuilder, Chicago, IL, USA] in 5μl of DMEM/hexadi- 111 methrine bromide (8mg/ml) were injected into the bladder lumen of mice by micro-sur- 112 gery as already described" I need more detail of the genetic construct and molecular mechanism of transgenic bladder tumor model. Just citing References 23 and 9 is not adequate.
Author Response
The manuscript needs the map and definition of the vector "The p53fl/flptenfl/flLuc-SIY strain results from a crossing between a p53fl/flptenfl/fl mouse 106 [9] with a mice carrying a transgene encoding for a T cell antigen (SIY) fused with the 107 luciferase (Luc) (B6.129S4-Gt(ROSA)26Sortm3(CAG-luc)Tyj/J, Jackson Laboratories). To 108 induce the deletion of p53 and pten genes and induce the expression of the Luc-SIY fused 109 protein into the bladder, 2.5x108 plaque-forming units of Cre-expressing adenoviral vector 110 [#AVL(VB181004-1095pzc)-K1, VectorBuilder, Chicago, IL, USA] in 5μl of DMEM/hexadi- 111 methrine bromide (8mg/ml) were injected into the bladder lumen of mice by micro-sur- 112 gery as already described" I need more detail of the genetic construct and molecular mechanism of transgenic bladder tumor model. Just citing References 23 and 9 is not adequate.
We apologize for the lack of clarity. We added a figure (Supplemental Figure 1) and detailed the adenovirus used (lines 116-123):
“To induce the deletion of the floxed genes, tp53 and pten, and the floxed STOP-cassette which inhibits the expression of the Luc-SIY fused protein (Supplemental Figure 1.A), a non-replicative adenovirus that expressed the Cre recombinase under the control of a CMV promoter (Supplemental Figure 1.B) is used [#AVL(VB181004-1095pzc)-K1, VectorBuilder, Chicago, IL, USA]. To induce bladder-specific deletion, 2.5x108 plaque-forming units of Cre-expressing adenoviral vector in 5μl of DMEM/hexadimethrine bromide (8mg/ml) were injected into the bladder lumen of mice by micro-surgery as already described [19].”
Reviewer 4 Report
Comments and Suggestions for Authors
"Development of Traceable Mouse Models for Advanced Bladder Cancer"
Title and Abstract:
- Title: The title should be clear and descriptive of its subject matter; accurately reflecting its content.
- Abstract: The abstract provides an informative summary of the study; however, more specific and detailed discussion regarding key results and implications would add value. Specifically mentioning success rate of metastasis development as well as resistance of models to anti-PD-1 therapy could enhance its value further.
Introduction:
The introduction effectively sets the scene for this study by outlining its significance and need for advanced mouse models. While background information may be extensive, consider including more recent references that reflect recent advancements in this field.
- Transition from introduction to research objectives should be seamless. Indicate which gaps this study seeks to fill.
Materials and Methods:
This section presents clear and organized methods. However, more could be accomplished by cutting back on redundant information; for example summarizing preparation steps for various reagents and buffers can make the section more concise.
- Ethic Statement: Check that the ethics statement is complete and includes exact dates and numbers for approval of its content.
Inducible Mouse Models: Determine whether any specific criteria were used when selecting mice as subjects as well as any exclusions that may exist.
Bioluminescence Imaging: Please specify which software version was used for image analysis to increase reproducibility.
- Statistical Analysis: Although software and significance levels have been mentioned, specifying which statistical tests were used in each analysis would be beneficial beyond what has already been noted in figure legends.
Results:
Luc-SIY Transgene and Tumor Development: Although comprehensive, this section could benefit from including more extensive comparison of tumor progression timelines across models.
Immune Microenvironment: While your findings are well-described, consider providing more context by explaining how they compare with previous studies or expected outcomes.
Resistance to Anti-PD-1 Therapy: This finding is of vital importance and could open up further discussion regarding possible mechanisms behind its observable resistance.
Immunogenicity of Luc-SIY Tumors: This section is excellent, using appropriate controls and including potential limitations or alternative interpretations of findings as appropriate. A brief discussion regarding potential limitations or alternate interpretations would also be beneficial.
Discussion:
The Discussion effectively interprets findings within existing literature and provides context. Expanding upon its novelty and significance of syngeneic orthotopic mouse model would enhance further.
Consider discussing the translational impact of these models for preclinical testing of new therapies, and their limitations are briefly mentioned here; expanding upon this and suggesting future research directions would offer a more balanced perspective.
Conclusion:
The conclusion provides a comprehensive summary of key findings; however, further elaboration on how these models could be utilized in future research and therapeutic development would be helpful.
Overall, this manuscript represents significant advances in developing traceable mouse models of advanced bladder cancer. With minor revisions and clarifications, it would make an invaluable contribution to this field.
Comments on the Quality of English LanguageThe manuscript is well-written, but a final proofreading for minor grammatical errors and typos is recommended.
Author Response
"Development of Traceable Mouse Models for Advanced Bladder Cancer"
Title and Abstract:
- Title: The title should be clear and descriptive of its subject matter; accurately reflecting its content.
We have slightly modified the title to: “Development of traceable mouse models of advanced and metastatic bladder cancer”
- Abstract: The abstract provides an informative summary of the study; however, more specific and detailed discussion regarding key results and implications would add value. Specifically mentioning success rate of metastasis development as well as resistance of models to anti-PD-1 therapy could enhance its value further.
We have modified the abstract to add details and discuss the values of the study (lines 28-36).
Introduction:
The introduction effectively sets the scene for this study by outlining its significance and need for advanced mouse models. While background information may be extensive, consider including more recent references that reflect recent advancements in this field.
- Transition from introduction to research objectives should be seamless. Indicate which gaps this study seeks to fill.
Recent publications have been added to the introduction (lines 85), the introduction was modified to make the transition from introduction to research objectives more fluid (lines 69-103) and objectives are more detailed (lines 95-103).
Materials and Methods:
This section presents clear and organized methods. However, more could be accomplished by cutting back on redundant information; for example summarizing preparation steps for various reagents and buffers can make the section more concise.
We summarized preparation steps and made the section as concise as possible without removing the important details for reproducibility of the study as suggested.
- Ethic Statement: Check that the ethics statement is complete and includes exact dates and numbers for approval of its content.
We added the dates of validation of the different authorizations.
Inducible Mouse Models: Determine whether any specific criteria were used when selecting mice as subjects as well as any exclusions that may exist.
We added the criteria used to select the mice for the inducible model (line 115):
“Male and female mice from 7 to 9 weeks old were used.”
Bioluminescence Imaging: Please specify which software version was used for image analysis to increase reproducibility.
We added the version of the Living Image software (line 158):
“Living Image© in vivo imaging software 4.7.3 (IVIS Imaging Systems).”
- Statistical Analysis: Although software and significance levels have been mentioned, specifying which statistical tests were used in each analysis would be beneficial beyond what has already been noted in figure legends.
We specified the procedure used for statistical analysis in this section (lines 220-225):
“GraphPad Prism 10 software was used for graph generation and biostatistical analyses. One-way ANOVA and two-ways ANOVA were first used for multiple group comparison and time follow-up, respectively. Then, specific statistical tests used for each specific experiment are detailed in the legend of the corresponding figure. For non-significant differences, p values are absent, while statistically significant results, p values are added on the figures: *p<0.05;**p<0.01;***p<0.001;****p<0.0001.”
Results:
Luc-SIY Transgene and Tumor Development: Although comprehensive, this section could benefit from including more extensive comparison of tumor progression timelines across models.
We added the bioluminescence quantification over time of the two genotypes (Figure 1C) to include more comparison.
Immune Microenvironment: While your findings are well-described, consider providing more context by explaining how they compare with previous studies or expected outcomes.
We provided details about the context and what we expected (lines 263-279).
Resistance to Anti-PD-1 Therapy: This finding is of vital importance and could open up further discussion regarding possible mechanisms behind its observable resistance.
We added this in the discussion part (lines 424-429).
Immunogenicity of Luc-SIY Tumors: This section is excellent, using appropriate controls and including potential limitations or alternative interpretations of findings as appropriate. A brief discussion regarding potential limitations or alternate interpretations would also be beneficial.
We discuss already this (lines 430-436). We added some limitations of our model lines 463-474.
Discussion:
The Discussion effectively interprets findings within existing literature and provides context. Expanding upon its novelty and significance of syngeneic orthotopic mouse model would enhance further.
Consider discussing the translational impact of these models for preclinical testing of new therapies, and their limitations are briefly mentioned here; expanding upon this and suggesting future research directions would offer a more balanced perspective.
We discussed in more details the advantages and disadvantages of our syngeneic model (lines 463-474):
“Even if syngeneic models allow the development of tumors in immunocompetent mice and are essential for the evaluation of immunotherapies [29], injecting a large number of tumor cells is very different from the natural evolution of a cancer, inducing some inflammation, limiting the clonal evolution of tumor cells and by consequence the intratumor heterogeneity. Our model of metastatic BC does not recapitulate the large TMB and the genomic heterogeneity found across BC patients. Otherwise, this syngeneic model of metastatic BC offers several advantages compared to the models already avail-able: the incorporation of the luciferase allows the detection of distant metastasis before necropsy; and this model is distinguished by its rapidity, simplicity of execution, and re-producibility, making it an invaluable resource for therapeutic studies. It also offers the possibility to perform genetic manipulations in tumor cells to understand BC's complex biology during its evolution, response to treatment or metastatic process.”
Conclusion:
The conclusion provides a comprehensive summary of key findings; however, further elaboration on how these models could be utilized in future research and therapeutic development would be helpful.
We added a sentence in that direction (line 473-474), just before the general conclusion.
Overall, this manuscript represents significant advances in developing traceable mouse models of advanced bladder cancer. With minor revisions and clarifications, it would make an invaluable contribution to this field.
Comments on the Quality of English Language
The manuscript is well-written, but a final proofreading for minor grammatical errors and typos is recommended.
We thank the reviewer about the several comments on the different parts that helped to improve our manuscript.